# Longevity, Centenarians and Modified Cellular Proteodynamics

**DOI:** 10.3390/ijms24032888

**Published:** 2023-02-02

**Authors:** Natalia Frankowska, Ewa Bryl, Tamas Fulop, Jacek M. Witkowski

**Affiliations:** 1Department of Physiopathology, Medical University of Gdansk, 80-211 Gdansk, Poland; 2Department of Pathology and Experimental Rheumatology, Medical University of Gdansk, 80-211 Gdansk, Poland; 3Research Center on Aging, Geriatric Division, Department of Medicine, Faculty of Medicine and Health Sciences, Université de Sherbrooke, Sherbrooke, QC J1K 2R1, Canada

**Keywords:** centenarians, longevity, proteodynamics, proteostasis, autophagy, geroproteome

## Abstract

We have shown before that at least one intracellular proteolytic system seems to be at least as abundant in the peripheral blood lymphocytes of centenarians as in the same cells of young individuals (with the cells of the elderly population showing a significant dip compared to both young and centenarian cohorts). Despite scarce published data, in this review, we tried to answer the question how do different types of cells of longevous people—nonagenarians to (semi)supercentenarians—maintain the quality and quantity of their structural and functional proteins? Specifically, we asked if more robust proteodynamics participate in longevity. We hypothesized that at least some factors controlling the maintenance of cellular proteomes in centenarians will remain at the “young” level (just performing better than in the average elderly). In our quest, we considered multiple aspects of cellular protein maintenance (proteodynamics), including the quality of transcribed DNA, its epigenetic changes, fidelity and quantitative features of transcription of both mRNA and noncoding RNAs, the process of translation, posttranslational modifications leading to maturation and functionalization of nascent proteins, and, finally, multiple facets of the process of elimination of misfolded, aggregated, and otherwise dysfunctional proteins (autophagy). We also included the status of mitochondria, especially production of ATP necessary for protein synthesis and maintenance. We found that with the exception of the latter and of chaperone function, practically all of the considered aspects did show better performance in centenarians than in the average elderly, and most of them approached the levels/activities seen in the cells of young individuals.

## 1. Introduction

In recent decades, lifespan has considerably increased and, at the same time, the worldwide population of centenarians has also enlarged. These days, centenarians’ prevalence rate in Europe is around 1 per 5000 and, nowadays, centenarians pose as an established model of successful aging [1]. It is justified by the fact that they can compress morbidity and comply with the group of individuals less prone to age-related diseases, including type 2 diabetes mellitus, neurodegenerative diseases (Alzheimer’s and Parkinson’s), cardiovascular disease, and cancers [2]. They have an intrinsic ability to relatively successfully react to stress factors in their daily life, described by the terms *resilience* or *hormesis*. Studies have also shown that centenarians are characterized by several special features. It is common knowledge that genetic factors may contribute to longevity. For instance, anti-apoptotic factors or pluripotency genes were found to be overexpressed in centenarians [3,4]. In another study, results suggest that centenarians could be less prone to oxidative stress owing to better-developed antioxidant mechanisms [5]. Identical observations were made for inflammaging and immune changes [1,6,7,8,9,10,11,12,13,14,15,16,17]. What is more, the endocrine system of centenarians is possibly capable of efficient glucose handling and shows higher insulin sensitivity compared to the young cohort [18].

It is a truism that proteins are the essence of life for every living creature. In humans, the estimated number of “own” structural and functional proteins reaches or even exceeds a staggering 100,000 [19]. The numbers of proteins provided by our mostly commensal microbiota are orders of magnitude larger. We recently proposed an overarching term of “proteodynamics” to include all facets of protein fates in our organisms. These facets start with obvious necessities, such as transcription of DNA, included in the genes onto different forms of RNA (notably mRNA, rRNA, tRNA, and many different species of short noncoding RNAs, including the regulatory microRNA, all modulated by the epigenome for an efficient transcription fulfilling its purpose), translation of nucleic acid chains onto amino acid chains of peptides, and their posttranslational modifications [20,21]. Then, matured proteins forming the proteomes are only functional for some time, undergoing multiple adverse changes in any living cell, including those related to oxidative stress, glycation, aggregation, and misfolding [21]. Eventually, in order for a cell to maintain some level of functionality for any longer period, these dysfunctional proteins should be identified and (proteolytically) degraded in a process called proteostasis [2,21]. With aging, the proteomes change both qualitatively and quantitatively, which led us to proposing the term “geroproteome” to reflect this fact. The process of aging-related changing of proteomes to geroproteomes leading to stepwise reductions in protein amounts and functionalities in aging cells is called “proteostenosis” [2,21,22].

It is a reasonable assumption that centenarians (and, in general, longevous people) might live that long, at least in part, due to better proteodynamics, i.e., handling of proteins from their generation to ultimate proteolytic degradation. However, published data on protein homeostasis/proteostasis/proteodynamics in longevous humans (centenarians to supercentenarians) are, so far, limited to a few papers. In our review, despite the scarcity of data, we describe the functioning of proteodynamics in centenarians and, more specifically, how its various components work. We will compare centenarian protein maintenance with that of younger cohorts in the search for potentially robust features of proteodynamics in the former, potentially participating in their extreme longevity.

## 2. The Question of DNA Maintenance

It is another truism that the quality of the proteome depends (in part) on the quality of the genome. Issues with genome maintenance are suggested as major risk factors of aging-related deterioration and diseases [23]. Thus, the first issue concerns the quality of centenarian protein-coding DNA and if the genes that regulate the maintenance of DNA may play a role in longevity. Not many published data indicate such a possibility and they are mentioned below. 

It was recognized at the beginning of this century that centenarians may have preferential expression of certain alleles, the products of which are important in DNA maintenance, e.g., BRCA1 [24]. Interestingly, in the same paper, we found data on a lower frequency of the APOE4 allele in centenarians compared to younger controls, which, considering that APOE4 is associated with a higher risk of both vascular dementia and Alzheimer’s disease, may also increase the survival of these individuals.

More recently, it was shown that at least a significant proportion of German and French centenarians expresses the exonuclease 1 (EXO1) gene, characterized by a functionally relevant SNP rs1776180. EXO1 plays important roles in human DNA repair [25] and the mentioned SNP leads to higher expression (and likely activity) of the product [26,27]. 

Then, genotyping of Ashkenazi Jewish population of 390 centenarians and 410 controls led to a discovery of the significant, more than 5-fold enrichment of the p.Y318C variant of the PMS2 gene coding for a mismatch repair endonuclease.

Eventually, a paper concerning analysis of 120 genome maintenance (GM) genes demonstrated, in addition to the already-mentioned EXO1 and PMS2, suggested that the following ones were strongly associated with longevity: Ku80 (non-homologous end joining), RAD51L1,3 and RAD52 (homologous recombination repair), NTHL1 and POLβ (base excision repair), and RAD23b (nucleotide excision repair) [23]. Interestingly, there is also a study showing that a specific decrease in the expression of XPD, a DNA helicase participating in nucleotide excision repair, is associated with longevity [28]. 

Furthermore, two studies concerning Italian and Chinese centenarians have shown enrichment of an rs189037 CT SNP in their ATM gene [29,30]. Interestingly, lymphoid cell lines derived from centenarians expressed significantly more ATM than those obtained from younger individuals [31].

Increased genomic stability of centenarian DNA may also depend on upregulation of the enzyme RNAseH2 in their fibroblasts and other (esp. myeloid) cells. Activity of this enzyme decreases the amount of cytoplasmic RNA:DNA hybrids (accumulating with age) which, in turn, reduces the amounts of IL-6 and type 1 interferon beta decreasing the level of inflammaging. Interestingly, the RNAseH2C locus is hypo-methylated in centenarian fibroblasts, while it becomes hypermethylated in senescent cells [32]. 

One of the DNA regions of intrinsically lower stability is rDNA coding the ribosomal RNAs. It was recently proposed that the instability of rDNA is one of the major sources of cytoplasmic DNA and RNA:DNA hybrids mentioned above, contributing to inflammaging [32]. Obviously, rDNA instability may result in changed output and quality of ribosomal RNA (and, eventually, the ribosomes); rDNA stabilization (as part of overall higher stability of DNA in centenarians) would clearly contribute to better proteome maintenance. 

Concluding, centenarians may be equipped better than people who would not reach that advanced age in the mechanisms mending the damage in their DNA and so have fewer adverse effects from age on genomes and, consequently, their cells may be less prone to proteostenosis due to such damage.

## 3. Transcription

### 3.1. mRNA: Quality (Transcription Fidelity) and Quantity

The obvious next step in the proteodynamics processes is transcription. Aging is generally associated with changes in mRNA levels, which may be due to decreased transcription, reduced mRNA stability, or both. It was also shown that centenarians show an mRNA expression profile more like young people than septuagenarians [33,34].

There are numerous studies showing a decrease or increase in transcription of specific gene(s) in centenarians as compared with younger individuals. However, they do not form a specific pattern and rather reflect adaptation of cells of a longevous person to long-lasting environmental and other (oxidative, ER, inflammatory, etc.) stresses.

The effectiveness of transcription depends, in part, on epigenetic modification (mainly methylation) of DNA. It has been postulated for more than a decade that unfavorable epigenetic changes (affecting the availability of certain stretches of DNA for transcription) may be one of the causes of aging [35]. They may also adversely affect the fidelity of transcription [36]. Thus, epigenetic changes in aging cells seem to result in global loss of the DNA methylation state [37]. An early study suggests that this hypomethylation may occur in a quasi-linear fashion, with it being lowest in newborn DNA, higher in the middle-aged, and highest in centenarian DNA [38,39]. Other works from the same period point out that although centenarians do indeed have lower global DNA methylation (and methylation of specific Alu sequences) than young individuals, their offspring retains “youthful” levels of DNA methylation when compared to the offspring of people who died between the 67th and 81st year of age. This may suggest the role of slower DNA demethylation in longevity [40,41]. On the other hand, a more recent study comparing DNA methylation patterns in small groups of Chinese centenarians and middle-aged controls demonstrated that genes associated with certain aging-related diseases, including type 2 diabetes, Alzheimer’s disease, and cardiovascular disease, were hypermethylated in longevous individuals (likely decreasing the risk of development of these diseases in studied centenarians [42]). A more robust Italian study using the concept of epigenetic clock demonstrated that both centenarians and their offspring retain younger DNA methylation patterns (their epigenetic clock is ticking more slowly) than offspring of people who never reached a hundred years of age [43,44]. Interestingly, analysis of the methylomes of peripheral blood CD4+ T cells from newborns, middle-aged people, and centenarians demonstrated more than 12 thousand differently methylated DNA regions between middle-aged people and centenarians, of which 4809 were hypermethylated and 7553 hypomethylated [45]. Further, newer papers conclude that the epigenetic age of centenarians is generally young compared with their chronological age [46,47,48]. There is a necessary caveat here: most of the papers cited above were studying changes in DNA methylation in the peripheral blood cells. Although it is very well known that the immune system aging impacts on aging of the whole organism and on the occurrence of aging-related diseases, the changes in methylomes in other cells (e.g., neurons, stem cells, or hepatocytes) may also impact on aging and longevity and such studies would be sought for. Concluding, epigenetics is recognized as an important factor in both human aging and longevity, which may directly affect the status of the proteome in the aged [49].

### 3.2. Noncoding RNAs: miRNA and lncRNA

Children of centenarians seem to share their parents’ mRNA and miRNA expression patterns (and, in this respect, are different from children of non-centenarians, which clearly suggests the genetic background for the trait [50]). Further, centenarians express some species of miRNA (notably miR-21) at the levels similar to young individuals and significantly different from those seen in septua- and octogenarians [51].

Moreover, the long noncoding RNA (lncRNA) species of centenarians may contribute to their extreme longevity in many ways, by regulating transcription and other processes. Thus, in a recent study, at least 11 lncRNAs were found to be overexpressed in centenarians compared to the younger (elderly) cohorts; 8 of them were identified as candidate healthy-aging-/longevity-related lncRNAs. It was shown that overexpression of two of these lncRNAs in centenarians, namely THBS1-IT1 and THBS1-AS1, is associated with decreased expression of cyclin inhibitors p16INK4 and p21cip, as well as with decreased activity of the senescence-associated β-galactosidase accumulating in senescent cells, suggesting the involvement of these lncRNAs in delaying cellular senescence in centenarians [52].

### 3.3. Posttranscriptional RNA Modification

Another process that may influence cellular proteodynamics is posttranscriptional modification of various types of RNA, including mRNA, tRNA, rRNA, as well as miRNA and other noncoding RNA species. Main chemical modifications of mRNAs are the N6 and N1 methylation of adenosine (m6A and m1A, respectively), C5 methylation of cytosine (m5C), hydroxylation of ribose, isomerization of uridine to form 5-ribosyl-uracil or pseudouridine, conversion of adenosine to inosine, and scores of other [53]. These modifications may affect nuclear export, transcript stability, splicing, folding, and localization, as well as translation initiation and fidelity, clearly influencing both quantitative and qualitative features of proteins, being the result of transcription of such modified mRNAs [54]. Among the cellular aging-related processes affected by RNA modification (mainly m6A and m5C methylation) are cell senescence, autophagy, inflammation, oxidative stress, and DNA damage [55]. Notably, m6A modification of relevant mRNA by METTL3 and METTL14 increases the production of cycle inhibitor p21, while m5C modification increases p21 and p16; accumulation of these proteins is characteristic for senescent cells. On the other hand, m5C modification of mRNA for p27, CDK1, and CDC25 by NSUN2 reduced senescence of human peripheral blood mononuclear cells [56]. However, activity of the same NSUN2 promotes senescence in HUVEC cells by affecting the expression of the SHC adapter protein family involved in RAS signaling [57]. 

Modification of noncoding RNAs (miRNA and lncRNA species), including, notably, both m6A and m5C, occurs thanks to the same molecular tools, i.e., methyltransferases and demethylases that participate in modifications of mRNA. Modification of miRNAs (m6A, as well as adenosine to inosine edition) changes their stability and decreases ability to pair with target mRNA sequences and to suppress translation, which may directly affect the efficiency of the translation process.

By far the most heavily and frequently modified RNA in mammals is the tRNA; it is calculated that one in five nucleotides in each tRNA molecule is modified, by methylation of adenosine, guanosine, cytosine, inosine, uridine and pseudouridine, acetylation of cytidine, and many others. These modifications occur in the anticodon loop, directly affecting the recognition of relevant codons and translation fidelity, or outside of these loops, affecting the structure of tRNA and, likely, its ability to bind relevant amino acids. 

On the other hand, posttranscriptional modifications of rRNA are scarce and involve no more than 2% of this RNA species. Not much is known about their effect on the functions of ribosomes; still, if they occur, it happens mainly in sites for tRNA binding, mRNA binding, and the peptidyl transfer center, suggesting that they may also be important for the maintenance of proteodynamics. 

It was found that RNA modifications may increase risks of various aging-related diseases, including AD and other neurodegenerative diseases, cancers, diabetes mellitus as well as atherosclerosis and its consequences—cardiovascular diseases and stroke, cataracts, and osteoporosis. Thus, decreased levels of m6A-modified RNAs are due to decreased function of methyltransferase METTL3/1 and affect the insulin/IGF1-AKT-PDX1 pathway participating in the development of T2DM [58]. Reduced activity of the same methyltransferase decreased neuronal levels of m6A in cells of AD patients [53]. It was also observed that oxidative stress increased m6A levels catalyzed by methyltransferase METTL14 [59]. This methyltransferase also performs the m6A modification of miR-19a miRNA, increasing its maturation; this, in turn, leads to increased proliferation of endothelia and may be involved in the development of atherosclerosis and its complications [60]. Then, activity of FTO demethylase is promoting the growth of non-small-cell lung cancer by regulating the levels of m6A modification [61]. The matter is complicated and requires further studies, as, for example, in AD, decreased activity of METTL3 is associated, on one hand, with a decrease in m6A modification of multiple transcripts, leading to decreased amounts of relevant proteins, but, on the other hand, it may lead to an actual increase in m6A modification of some RNAs [53,62,63]. 

Thus, RNA modifications may shorten life (prevent longevity) by promoting the development of aging-related diseases and it is, therefore, feasible that (opposite?) RNA modifications should promote human longevity. However, until now, there are virtually no published data on RNA modification in centenarians and its role for effective maintenance of their proteomes.

## 4. Translation: Effectiveness and Fidelity

A quarter century ago, it was recognized that, with age, the number of nucleoli (intranuclear centers of rRNA production) decreases, which may result in a lower yield of ribosomes and also result in decreased overall production of proteins in the cells of old individuals. However, the paper describing this study concerned an 87-year-old participant as the oldest subject in the study (as all in this study, tested twice, at final age of 92), so it did not give any insight in the ribosomal production and quality in centenarians [64]. 

Some papers hint at the decreased quality of ribosomal proteins in the aged (not human) organisms. The first one concerns the aging *C. elegans*; it was found that a significant proportion of ribosomal proteins of aged nematodes becomes SDS insoluble, which likely depends on changes in genes for relevant proteins [65]. However, recent studies suggest that, in fact, it is decreased ribosomal production and associated translation of ribosomal and other proteins that occur with aging [66]. A recent review highlights the notion that ribosomal biogenesis is, in fact, one of the most energy-consuming cellular processes, in some cells utilizing up to 80% of available energy [67]. As is well documented, increased energy expenditure shortens cellular life, and it is postulated that slowing down or halting of this energy expenditure (including on ribosomal protein (RP) transcription and translation) may be a common mechanism for extending lifespan [67]. Thus, aged cells have fewer ribosomes with poorer RP quality, which results in lower output of new proteins. 

Ribosome production depends majorly on the mTOR and sirtuin pathway, which seems to link the life-extending effects of mTOR inhibition (and sirtuin stimulation) with ribosomal reduction. SIRT1 expression was found to be increased more than two-fold in centenarian lymphocytes compared with those from middle-aged to old individuals [68]. According to Blagosklonny, “certain genetic variants such as hyper-active mTOR (mTarget of Rapamycin) may increase survival early in life at the expense of accelerated aging”, which can be understood as a form of the antagonistic pleiotropy effect. This means that a maintained highly active mTOR with aging would limit the individual’s survival. In the past, robust individuals with high mTOR activity predominated, but died earlier (and so, fewer centenarians were around); now, with the help of better life conditions and contemporary medicine, less robust people (who would die without these helping factors) survive longer, contributing to the generation of a growing cohort of humans with low mTOR pathway activity to become centenarians [69,70].

One needs to be aware that mTOR is much more than a single-purpose protein or pathway, related only to the sirtuin pathway, as mentioned above; rather, it serves as a master regulator (hub) of cellular metabolism. Multiple processes affected (regulated) by the mTOR include mRNA translation, protein biosynthesis, mitochondrial function and biogenesis, stress responses, and autophagy, making it an extremely relevant factor when considering longevity and proteome maintenance. Because of this, an increased amount and activity of mTOR signaling is included among the nine hallmarks of aging [71]. It was reported that mTOR signaling is reduced in longevous mammals and humans [72]. As the downstream processes regulated by the mTOR network are dependent on nutrient availability (which itself may change with aging due to many factors), they are an indirect subject of caloric restriction—a successful intervention, decreasing rates of aging. 

Apparently, the properties of transfer RNA (tRNA) are important in aging, although available data are derived mostly from studies in model organisms. Aging seems to affect tRNA stability, recognition of codons, and binding to relevant amino acids (aminoacylation). This may be related to the extreme modifiability of tRNA [73]. Studies on human fibroblasts have demonstrated that the changes in the activity of the NSun2 tRNA methyltransferase involved in tRNA modification may lead to postponement of fibroblast senescence, by inhibiting expression of cyclin kinase inhibitor p27^KIP1^ or its acceleration [74]. On the other hand, activity of NSun2 actually accelerates stress-related senescence of HUVEC cells [57]. It was also found that silencing of the methyltransferase, called Dnmt2 in the same model cells, induces their senescence [75]. It may be of interest that another member of the NSun methyltransferases’ family, NSun5, was found to affect longevity in yeast, *C. elegans*, and fruit flies. The enzyme is known to methylate a specific residue (C2278) within the conserved region of 25S rRNA rather than tRNA. Loss of the NSun5 homolog in yeast (Rcm1) results in changes in structural conformation of ribosomes and loss of translation fidelity; on the other hand, changes in the ribosomes favor recruitment of oxidative-stress-responsive mRNAs into polysomes and eventually increase the lifespan of studied species [76]. So far, there are no such data coming from studies in mammals.

All these data suggest that the effect of the mentioned methyltransferases on aging is not directly associated with modification of tRNA properties in the aged. The only, very recent paper dealing with the question about possible modifications to tRNA in centenarians was actually studying the correlations between nutrition and longevity in a Chinese population. The authors found that compared to younger cohorts, concentrations of centenarian plasma metabolites, including VLDL, lactate, alanine, NAG, citrate, tyrosine, choline, carnitine, TMAO, β-glucose, α-glucose, valine, and unsaturated lipids, correlated with longevity, on one hand, and were associated with enrichment of aminoacyl-tRNA biosynthesis on the other [77]. Another paper by the same group demonstrated that diet leading to reductions in cardiovascular risk affected the same pathway in centenarians [78]. This might suggest that the status of tRNA is important for living an extremely long life.

Directly related to the state of mRNAs, tRNAs, rRNAs, and other RNA species in centenarians is the question of translation fidelity. Errors in translation were proposed to be paramount for cellular aging within Orgel’s “error catastrophe” theory of aging in the 1960s [79]. Plausible as it was, the theory could not be adequately tested then due to the lack of sensitive-enough methods of detection of such errors, which are estimated to occur every 2000–10,000 amino acids and levels of mistranslated proteins in normal young cells are negligible; thus, the theory was neglected over the next few decades as baseless and ultimately all but forgotten [80]. Only in recent years was it skillfully dusted off. Thus, in 2013, Azpurua et al. compared translation error levels in the fibroblasts of mice and of longevous naked mole rats and found that the number of translation errors in mouse cells was up to an order of magnitude (10-times) higher than in the cells of mole rats [81]. These observations were expanded by Ke et al., who studied translation fidelity in 17 rodent species with different lifespans; these authors found that translation fidelity at the first and second codon positions positively correlates with species maximum lifespan. Therefore, more longevous species had fewer translation errors (higher translation fidelity) [82]. The authors are prudent to say that inter-species differences in translation fidelity (and associated shorter or longer average lifespans) do not translate to individual longevity or organisms with more accurate translation; however, they propose that decreased translation fidelity with aging may be partially responsible for age-related neurodegenerative diseases in humans, including Alzheimer’s, Parkinson’s, and Huntington’s diseases [80]. On the other hand, it was demonstrated that a relatively minor increase in the rate of translation errors may downregulate the synthesis rate of the affected protein, increase the protein aggregation, and activate the proteotoxic stress response in human cell lines, making studies of the effect of changes in translation fidelity for human longevity worthwhile [83,84].

A special case worth considering here is aging-related changes in translation fidelity and output of mitochondrial proteins. Thus, the authors of one recent paper suggest “tradeoffs” between protein synthesis fidelity and rate, which may affect the functioning of (aged) cells in the generally stressful environment and demonstrate that mice with mutation in the Mrps12 gene (possessing so called hyper-accurate (Mrps12^ha/ha^) mitochondrial ribosomes, i.e., performing fewer translation errors than wild-type animals) are actually characterized by reduced lifespan due to development of hypertrophic cardiomyopathy, which indicates that the issue of translation fidelity vs. longevity is complex and requires further studies [85].

## 5. Energy Requirement of Proteodynamics: Do Centenarians Stand Out from among Aged Individuals?

The last paragraph brought our attention to possible aging-associated modifications in mitochondrial proteins (mitoproteome), likely associated with changes in transcription and translation fidelity of mitochondrial proteins, whether manufactured based on mtDNA code or coded by nuclear DNA. Obvious consequences of issues with mitochondrial function in aged cells will be decreased output of ATP on one side and, likely, increased output of ROS. In fact, mitochondrial dysfunction is described as another of the nine hallmarks of cellular aging [71]. 

Protein manufacturing is one of the most energy-consuming processes in cells so it is obvious that any reduction in available energy would adversely affect protein output. 

Indeed, it was found in numerous works that the activity of multiple enzymes involved in oxidative phosphorylation (OXPHOS) significantly declines with age, which is associated with deletions in mitochondrial (mt)DNA [86,87]. This was shown to be directly associated with decreased mitochondrial ATP output and OXPHOS coupling in cells of aged individuals, although other papers put this observation in question [88]. For instance, a recent paper demonstrated significantly increased demand of centenarian muscles for ATP (in mM/W) during exercise, although muscle mitochondrial ATP synthesis capacity did not differ between young, elderly, and the oldest individuals [89]. A proposed explanation for these phenomena is that in the extremely old individuals, there is a hypertrophy of remaining muscular fibers accompanying general muscle atrophy (which clearly might be associated with decreased output of muscular proteins) [89].

Varying mitochondrial processes may be associated with the existence of multiple mtDNA variants, called haplogroups. It was found that at least one such haplogroup, designated as J, is over-represented in longevous people and centenarians from North Italy, Ireland, and Finland, but, surprisingly, no such relation was found for Southern Italians, which may suggest that the effect of over-representation of a specific mtDNA haplogroup may be population-specific and further studies on the matter are desirable [90,91,92,93,94]. 

It was demonstrated that decreased ATP output from centenarian mitochondria is associated with increased output of oxygen free radicals (ROS), at least in vitro [5]. The ROS are strongly and universally implicated in the process of cellular aging by leading to oxidative stress. The cited paper shows that cellular bioenergetic preservation in centenarians, despite increased ROS output, may be due to compensatory effects, including increased mitochondrial mass per cell. The latter results from mitochondrial hyperfusion (described as mitochondrial “hypertrophy”) leading to elongated shape and, more importantly, decreased mitophagy (a form of autophagy eliminating failing mitochondria) [5]. This would represent a perfect adaptation/resilience to repetitive stresses in centenarians [95]. On the other hand, moderately increased mitochondrial ROS (mtROS) output is considered beneficial, promoting longevity [95]. Accumulating ROS are a stressor and stimulate mitochondrial and other adaptive cellular stress responses. By analogy to well-known hormetic effects, where a small amount of stressor is beneficial and larger doses of the same stressor become harmful, the effect of some increase in mtROS is known as mitohormesis [95,96]. It was shown in model organisms (from C. elegans to fruit flies to mammals) to prolong life, and at least in some studies, the effect was nullified by antioxidant treatment. Although not yet demonstrated in centenarian cells, mitohormesis may conceivably play a role in extreme longevity. As mentioned above, the mTOR pathway regulates mitochondrial biogenesis and function, including the role of mitochondria in apoptosis, but also mitophagy and mitohormesis, so its effect on longevity may be exerted also via modification of mitochondria [95,96,97]. 

## 6. Effectiveness of Posttranslational Modifications

Most of the proteins are not just polypeptides, but in order to become functional, they must have attached multiple functional residues, including sugars, lipids, and other residues, added in the complex process of posttranslational modification of proteins. 

Thus, it was recently shown that the patterns of glycosylation of cellular proteins significantly differed in centenarians/semi-supercentenarians (SSCs) from both elderly and young individuals. Specifically, the increase observed in SSC concerned multi-branched and highly sialylated N-glycans. Interestingly, both of these glycan species are implicated in anti-inflammatory processes; the concurrent finding of elevated levels of CRP, IL-6, and TNFα in SSC suggests a fine, life-prolonging balance of pro- and anti-inflammatory processes in the oldest individuals [98]. One needs to stress here that, so far, there are no published reports on age-related changes in N-glycosylation effectiveness concerning intracellular proteins. The paper cited above deals with human plasma proteins, as well as two earlier papers demonstrating and discussing the role of decreased N-glycosylation in sera of elderly individuals [99,100]. On the other hand, glycans are present mostly at the cell (external) surfaces, where they serve as signaling molecules for (inter alia) intercellular communication, signal transduction, adhesion, and endocytosis [101]. All of these processes are affected in the elderly. A recent paper comparing multiple gene variants for metabolic pathways, including, notably, those associated with immunity and endocytosis, in almost 300 cognitively healthy centenarians, 1779 elderly, and 1630 Alzheimer’s disease patients, demonstrated that healthy centenarians had a significantly lower frequency of variants associated with elevated risk of AD and poor immune reactivity then both healthy elderly and AD patients [102]. Although these findings do not by themselves demonstrate more robust N-glycosylation (and its direct consequences) in healthy centenarians, the relationship seems very feasible. 

Lipidation of proteins includes cysteine prenylation, N-terminal glycine myristoylation, cysteine palmitoylation, and serine and lysine fatty acylation. Prenylated proteins may incorporate in cellular membranes and serve as e.g., signal transduction elements. Although protein prenylation is listed among very important modifications of multiple proteins, not much is known about its relation to aging and practically nothing about the course of this modification in human centenarians. One important finding associates impaired prenylation (elevated farnesylation) with increased risk of Alzheimer’s disease (and, obviously, earlier demise) [103]. In fact, it is supposedly decreased prenylation which prevents neurodegeneration, participating in multi-prong effects of statins [103]. Further, positive results of a clinical trial aiming at inhibition of prenylation in Hutchinson–Gilford progeria (where lamin A farnesylation is impaired) suggest the potentially important role of control over prenylation in longevity [104]. Still, such direct studies are, so far, missing.

The capacity of cells to alleviate oxidative or other damage to their DNA is one of the life-prolonging factors and it was found to decrease with age and be lower in short- than long-living species. One of posttranslational modifications of nuclear proteins in response to these stresses is polyADP-ribosylation performed by an enzyme poly(ADP-ribose) polymerase (PARP). Correlation between maximal PARP activity and species-specific lifespan has been demonstrated. Further, recent papers show that in centenarian cells, both the amount and the maximal activity of PARP were significantly higher than in elderly controls and similar to that seen in young individuals [105,106]. On the other hand, proinflammatory effects of PARP activity are mentioned as the opposite to its role in extending the cells’ life; this paper does not consider the centenarians and their high (albeit balanced) proinflammatory activity [107,108].

In line with the above, retained cognitive capacities in centenarians may depend upon yet another protein modification: carbonylation. It was recently found that the carbonylation of circulating proteins is the lowest in centenarians with highest cognitive capacities. As carbonylation is related to oxidative stress and DNA metabolism (manifested e.g., as neuron damage), its lower values in cognitively successful centenarians may have a causative role [109].

Other posttranslational protein modifications, including O-glycosylation, glycation, ubiquitination, nitration, SUMOylation, prolyl-isomerization, and truncation, were found to happen to the tau protein, possibly facilitating the production of neurofibrillary tangles (NFT) leading to neurodegenerative pathologies; still, the role of these processes for human aging and longevity is not known [110]. 

A special case of posttranslational modification is limited (regulatory) proteolysis of nascent peptides as well as of matured, functional proteins. For the proteins remaining in the cell (not being secreted), a group of neutral cysteine proteases called calpains is the main effector of this modification. Their behavior in aging cells will be described later.

## 7. Proteomic Signatures in Centenarians

A balanced cellular proteome, where necessary proteins are made in adequate quantities and retain functionality for a reasonably long time, only to be removed if defective, misfolded, etc., by appropriate proteolytic systems, is maintained by proteostasis, which controls the cycles of protein synthesis and degradation. Loss (at least partial) of proteostasis characterizes aging cells in various human tissues and organs, such as skin, muscle, lymphocytes, liver, lung, heart, kidney, and brain. It is considered a hallmark of cellular senescence [111]. 

According to new works, the centenarian proteomes significantly differ from those of the elderly, both qualitatively and quantitatively, suggesting that from the point of view of their proteins, centenarians are aging slower than the rest of the population [13,112,113,114]. The utility of plasma proteome in human aging/longevity studies was recently summarized in a review by Deutsch et al. [115].

### 7.1. Proteostatic Systems and Autophagy in Longevity

Apart from the multiple cellular processes associated with protein manufacturing and maintenance characterized above, there is another facet of the cellular protein fates: proteolytic degradation of used-up, spent, malformed, and aggregated proteins, known as proteostasis and included among the nine pillars (hallmarks) of cellular aging [71]. It is calculated that as much as 30 to 90% of all manufactured proteins contains sequence and/or structural errors, leading to their malfunction and early aggregation [116]. Elimination of these faulty proteins, which—if not removed—may adversely affect the cellular functions, would include means to detect these proteins and to direct them to the appropriate lytic machinery, and the cellular proteolytic machineries themselves. The cellular aging process is accompanied by the progression of malfunctioning protein quality control systems [117]. The overarching term for these processes of cellular protein quality control is autophagy; it consists mainly of the ubiquitin–proteasome system and of the lysosome pathway.

#### 7.1.1. Autophagy—The Ubiquitin-Proteasome System (UPS)

The ubiquitin–proteasome system stands as the main proteolytic system of degradation of misfolded, damaged, or oxidized proteins within all protein quality control systems [118]. The 20S proteasome constitutes four rings (with α and β subunits within rings) that divide into a total of 28 subunits. It exhibits proteolytic properties due to the presence of threonine proteases with caspase-, trypsin-, and chymotrypsin-like activities [119]. The 19S particle is an element of the proteasome that performs a regulatory role and its association with the 20S proteasome results in the formation of the 26S proteasome that allows for the degradation of ubiquitinated proteins in an ATP-dependent manner. The proteolytic system described briefly above takes part in many processes that, for example, include cell growth, DNA replication, or cell metabolism.

During lifetime, the cellular environment is prone to constant oxidation that negatively correlates with proteasome activity and increases the risk of accumulation of previously incorrectly modified proteins. It has been shown that irreversible oxidative modification of the proteasome complex impairs its activity [120,121]. Various other reasons, such as declined chymotryptic activity or inefficient expression of regulatory subunits, are also causes for that phenomenon [122,123]. 

Nevertheless, a study performed by Chondrogianni and her team provided evidence that fibroblast cultures from healthy centenarians have a sustained activity in the proteasomes [124]. RNA expression levels of both catalytic and regulatory subunits were checked and, in all cases, expression in the centenarian cultures was at a similar level to that observed in the young cohort, whereas the level of expression in the material from old individuals was significantly lower than the others. These results raised the question of whether the proteasome *activity* of centenarian cells would be maintained at a similar level compared to the young controls. In an attempt to investigate that, the team performed the peptidylglutamyl–peptide hydrolase activity assay and also checked for chymotrypsin-like activity in the proteasome. The results were consistent with a previous experiment performed at the molecular level. Fibroblast cells derived from centenarians represented a considerable proteolytic activity similar to that displayed by the cells of young individuals. What is more, the study of oxidized proteins within cultures derived from individuals of different ages suggests that higher activity and functionality of proteasomes correlate with decreased amounts of oxidized proteins. To summarize, centenarians’ functional proteasomes may contribute to the better-preserved protein turnover that would allow for more prosperous aging. Mouse studies of proteasome activity also show that extremely old individuals have preserved proteasome function, which also suggests that this is a feature of successful aging [125]. 

It is of interest that the process of ubiquitination (adding ubiquitin residues to doomed protein with the complex help of different enzymes, mainly ubiquitin ligases) did not amass so much attention regarding longevity as the function of proteasome itself. Ubiquitination was, however, studied in cells from Alzheimer’s disease (AD) patients, where it was shown to be greatly affected by changes in activities of multiple ubiquitinases and de-ubiquitinases, summarily leading to accumulation of such pathological peptides/proteins, such as amyloid-beta, phospho-Tau, and alpha-synuclein [126]. 

Although not studied much in longevous humans, it was recently found that overexpression of E3 ubiquitin ligase WWP-1 extends the lifespan of C. elegans in response to diet restriction [127]. The authors of the cited paper suggest that due to strong conservation/homology of the nematode’s wwp-1 with mouse and human WWP1 ligase, the role of its preserved or enhanced activity in increasing the longevity of the latter is conceivable. On the other hand, it was found earlier that caloric restriction prevents accumulation of ubiquitin and ubiquitinated proteins in aging Emory mice [128]; thus, the topic of a potential role for changes in protein ubiquitination for human longevity remains open for studies.

#### 7.1.2. Autophagy—Lysosome Pathway

In this process, unnecessary and faulty cytoplasmic material (aggregated proteins recognized and addressed to the lysosomes by various chaperone proteins, as well as larger structures including damaged organelles) is incorporated into lysosomes where, owing to the presence of a vast array of hydrolases, is then broken down into elementary compounds [129]. Therefore, autophagy contributes to the reduction in damaged proteins within the cellular environment and, at the same time, provides amino acids for anabolic processes and the synthesis of new pule of proteins. In contrast, dysfunction of autophagy observed in the cells of old individuals leaves behind lots of intracellular “garbage”, which, eventually, spilling from damaged or dead cells, stimulates innate immunity, participating in inflammaging [130]. Autophagy plays a prominent role in lymphocyte homeostasis and in the survival of effector T cells as well. It is also required for proper functioning of mitochondria and endoplasmic reticulum and is involved in lipid metabolism [131]. 

With aging, protein homeostasis starts to fail and the effectiveness of autophagy drops what has been documented in many cells and tissues, including cardiac myocytes, fibroblasts, or hepatocytes [132,133,134]. It has also been proposed that malfunctioning of the autophagic system with age can have negative implications regarding the immune system. In a study by Bektas et al., it was found that in the aging CD4+ T cells, the process of mitochondrial respiration was impaired due to faulty mitophagy. That resulted in numerous lymphocytes that contained non-digested but dysfunctional mitochondria. Such disrupted mitochondria turnover could finally lead to the triggering of chronic inflammatory diseases and impairment of adaptive immune defense among the elderly [135]. In another study, T lymphocytes in which basal levels of autophagy were decreased were isolated from old individuals. It led to the assumption that a decline in autophagy is associated with features characteristic of the senescent immune system [136].

Nevertheless, studies on centenarians focused on the functioning of their immune system, providing evidence that such immune systems can be well maintained and their immune responses are improved in a similar way as in young individuals. Thus, for example, NK cells in centenarians stand out with cytotoxic capability similar to the young cohort. Further, the expansion of CD4+ cells in supercentenarians has been found, while centenarian CD8+ cells differentiate with decreased expression of CD28+ and higher expression of CD45RA [137,138,139,140]. In another study, Raz et al. investigated autophagic activity in CD4+ T cells of centenarians and their offspring. The previously mentioned immune cells were analyzed in terms of measurement of autophagy flux, which served as a parameter to evaluate the autophagy activity that was previously induced. One of the results showed that induction of autophagy was severely compromised in helper T cells collected from the offspring of parents with usual survival compared to cells that were obtained from the offspring of parents with exceptional longevity and lymphocytes from young control as well. In the case of autophagic activity in centenarian T cells, it was at a similar level as cells from the progeny of subjects with typical survival, while centenarian offspring retain higher autophagic activities [141]. Such results have led to the assumption that an age-associated decrease in autophagy in the T-cell population cannot be fully avoided, only slowed down. Another study worth mentioning concerned transcriptome analysis of autophagy–lysosomal function in centenarians and their children. The finding showed that some of the genes encoding the autophagy–lysosomal pathway components were up-regulated as it may have been a way to preserve longevity [142]. Further, lysosomal contents and activity in immortalized centenarian B lymphocytes were shown to be higher than in the younger cohort, again stressing the role of efficient autophagy in extreme longevity [143].

Generally, increased autophagy is considered a pro-longevity process, which apparently converges multiple “longevity paradigms”, including reduced insulin/IGF-1 signaling, mTOR signaling, and mitochondrial respiration (OXPHOS), being controlled via multiple autophagy regulators, including HLH30/TFEB, MML-1/Mondo, ATG family, forkhead transcription factors (FOXO), and sirtuin 1 [144].

The mechanisms behind the above observations of preserved/maintained relatively high levels of cellular autophagy may include changes in the cellular chaperone systems, in their lysosomal contents and activities, or both.

A recent transcriptomic study revealed that the autophagy/lysosomal pathway was enhanced in centenarian cells compared with those of younger individuals, suggesting relatively robust cleaning of faulty proteins in the former [142]. Altogether, up to 38 genes providing transcripts important for autophagy and lysosome function are upregulated in centenarians, compared to their spouses; elevated expression of these genes is retained in the cells of centenarian offspring [142]. Expression, amount, and, in consequence, activity of an important lysosomal enzyme, N-acetyl-alpha-glucosaminidase (NAGLU, degrading heparan sulfate), is significantly increased in the cells of centenarians compared to younger individuals. Further, the serum levels of beclin-1, which regulates lysosome biogenesis and autophagy, are elevated in centenarians and their offspring, as compared, e.g., to centenarian spouses (eliminating environmental bias) [142,145].

Heat-shock proteins (HSPs) or chaperones, as mentioned above, detect aggregated or otherwise faulty proteins and guide them to the sites of intracellular degradation; they also participate in correct assembly and folding of nascent proteins. It was found that both the quantitative and functional deficit of chaperones is associated with aging [146]. Of special interest there is the Hsp70 chaperone, which is protective against the harmful effects of oxidative stress and a modulator of the inflammatory status and, thus, participates in the process of aging [147]. It was found that the levels of Hsp70 and of Hsp60 in nonagenarians and older individuals are significantly lower than in younger people [148]. A newer study, which included centenarians, did show that, in fact, low serum Hsp70 levels are associated with extreme longevity, and that they are inheritable by centenarian offspring [149]. On the other hand, another paper coming from roughly the same period states that heat-induced production of Hsp70 in the EBV-transformed B lymphocytes from centenarians is similar to that detected in the cells of young and adult individuals, which would correspond to the lysosomal pathway data cited above [150].

Concluding, we can say that centenarian cells seem to maintain robust autophagic mechanisms, which may help them to retain sufficient functionality, even at an advanced age [142,145].

#### 7.1.3. The Calpain–Calpastatin System

The calpain–calpastatin system also takes part in maintaining overall protein homeostasis. Calpains belong to the family of neutral cytosolic cysteine proteases, which are highly dependent on the defined concentration of calcium ions in the cellular environment in order to be activated [151,152]. When active, they are capable of proteolytic processing of substrates, resulting in their subsequent functional modification (activation or inactivation) via modification of their structure, or ultimately in their degradation. Calpains are in tight control by their natural endogenous inhibitor calpastatin. What is more, they have been proven to regulate at least 300 (and likely many more) cellular proteins of various types. These include biomolecules that are compounds of the apoptosis pathway, signal transduction pathway, membrane pumps, and channels in the immune as well as other body systems [21,153,154]. 

Our team has demonstrated that genes in the calpain–calpastatin system are constitutively expressed in the resting T cells and that calpain activity in these cells is necessary for their subsequent activation, synthesis of cytokines, and proliferation [154]. We also investigated the amounts and activities of ubiquitous calpains (calpain-1 and calpain-2) and calpastatin in the leukocytes of subjects of various age groups. In the CD4+ and CD8+ T cells and B lymphocytes of healthy elderly people, we observed decreased levels of amounts and activities of previously mentioned proteins when compared to a young cohort [155]. In another experiment, regarding lymphocytes of centenarians, although the activity of calpains did not differ significantly from cells obtained from the elderly, their amount was as abundant as in young individuals (in preparation). It is presumable that the immune system of centenarians invests in a constitutive expression of calpains as it is constantly in use in order to cope with possible external factors and also due to the vast number of substrates that play various roles in the cellular environment.

#### 7.1.4. ADAMS and ADAMTS

The final group of cellular proteases, consisting of the family of a disintegrin and metalloproteinases (ADAM) and of ADAM with thrombospondin domain (ADAMTS), is (somewhat similarly to calpains) involved in modification of multiple protein substrates associated with intra- and intercellular signaling (e.g., by facilitating shedding of soluble proteins from the cell membranes). It was found that activities of at least some of the members of the ADAM family (notably ADAM 13) are reduced in the cells of Japanese octogenarians compared to young adult and middle-aged individuals [156]. However, according to our knowledge, no published data exist yet concerning the ADAM or ADAMTS amounts and/or activities in centenarians.

## 8. Conclusions

Despite relatively little and certainly incomplete data, it can be concluded that the retained efficiency of proteolytic systems in addition to earlier processes of transcription, translation, and posttranslational modifications translates into proper proteodynamics and is associated with delayed aging and attaining longevity. As a result, the centenarian organism is able to respond effectively to external factors and maintain the internal balance of the proteome, while individuals who would never reach a hundred or more show increased modifications and flaws in their cellular proteomes, which become geroproteomes, with more and more limited functionalities [20,21]. A graphical summary of the described phenomena is shown in Figure 1. Data from the papers that we cited in this work suggest that robust young proteomes remain relatively untouched in (healthy) centenarians. To some extent, this phenomenon may be associated with genetic variants of scores of genes associated with multiple stages of proteodynamics, assuring, or at least participating in, longevity. On the other hand, we can speculate that healthy proteomes of centenarians are the result of their adaptative/hormetic/resilient reactions to stresses and insults, participating in their complex maintenance of robustness, leading to longevity. 

The maintenance of proteodynamics in centenarians is a perfect example that chronological age per se is not the cause of the decreased longevity and functionality, but the complex interplay of adaptation/resilience/hormesis with lifelong stresses define the successful aging represented by the centenarians. Still, more studies should be conducted in order to fully comprehend the role of maintenance of homeodynamics for extreme longevity and to—possibly—set targets for desired interventions.

## Figures and Tables

**Figure 1 ijms-24-02888-f001:**
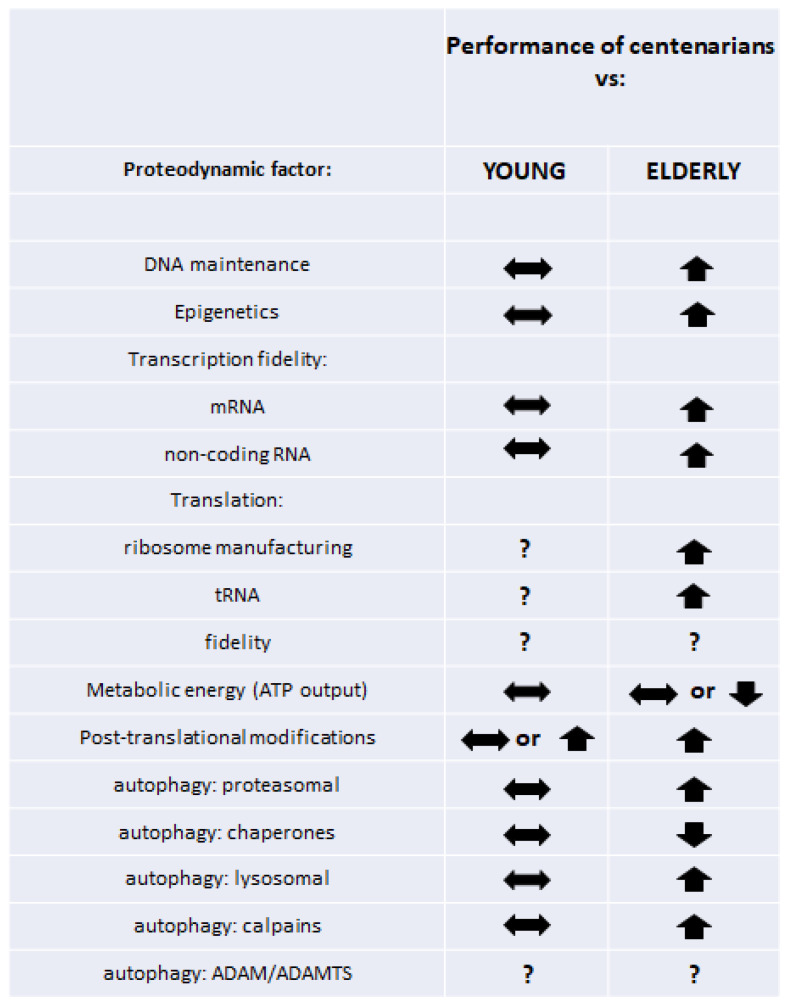
Comparison of performance of various facets of proteodynamics in the cells of centenarians versus young and elderly individuals. 
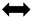
—no significant difference; 
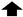
—better/higher than in compared age group; 
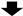
—worse/lower than in compared age group; **?**—no or inconclusive data.

## Data Availability

No new data were generated in this review.

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
