# Peer review of "Longevity, Centenarians and Modified Cellular Proteodynamics"

_ijms, 2023, doi:10.3390/ijms24032888_

Round 1

Reviewer 1 Report

The manuscript by Frankowska et al., is a narrative review on the many different aspects of proteostasis/proteodynamics of centenarians taken as a model of human longevity. The paper is of interest and the authors are expert in the field, however, there’s a number of points that can be addressed to improve the review and make it more suitable for publication.

Abstract: it is little informative and it should be rewritten describing what are the actual results instead of saying what the authors will do in the main text, and what is the hypothesis that they put forward.

Introduction: the concept of proteostasis is introduced twice, please harmonize the text.

Chapter 2: As far as the role of DNA repair in the aging process, other theories can be proposed, i.e. some DNA repair mechanisms can be associated with inflammation. See and discuss the data by Storci et al., 2019 and 2020 (PMID: 30622304; PMID: 31926964).

Chapter 3.1: I will add a caveat on the tissue type where the DNA methylation age has been measured (usually blood cells), which does not necessarily reflect the trend of other cell types that may be more relevant for longevity (i.e. brain, liver, stem cells).

Chapter 3.2: Ref. n. 51 is not correctly reported, please rephrase.

Chapter 4: the part on mTOR is oversimplified, since mTOR is not merely a protein whose function can be decreased or increased, but it is rather a hub where an integration of signals occurs leading to a decision on cell fate. Moreover, from an evolutive point of view, if the authors were right, it is more than likely that hypofunctional variants of mTOR would have been eliminated by natural selection. Please reconsider this section.

Chapter 5: please explain more in detail the hypothesis of mitohormesis. A brief mention should also be done to the fact that a little mitochondrial dysfunction can be good for longevity as it can elicit a beneficial stress response.

Chapter 6: a more detailed discussion can be provided for N-glycosylation (e.g. PMID: 22353383; PMID: 18047421).

Chapter 8: an authors’ expert point of view would be desirable. Would the authors mind to speculate on the geroproteome? What are the causes for its onset? Have centenarians always been exceptionally gifted in terms of proteome maintenance, or they adapted better than others to environmental situations? At least in hypothetical form, a guess on this would be welcome.

Minor points: line 46: a series of 3 full stops is present; Chapter 8: the word “conclusions” contains a typo.  

Author Response

Abstract: it is little informative and it should be rewritten describing what are the actual results instead of saying what the authors will do in the main text, and what is the hypothesis that they put forward.

         We thank the reviewer for this remark. We have expanded the abstract in order to incorporate requested information (added lines 17-28).

Introduction: the concept of proteostasis is introduced twice, please harmonize the text.

       We thank the reviewer for pointing this out. Repetition has been removed.

Chapter 2: As far as the role of DNA repair in the aging process, other theories can be proposed, i.e. some DNA repair mechanisms can be associated with inflammation. See and discuss the data by Storci et al., 2019 and 2020 (PMID: 30622304; PMID: 31926964).

         We thank the reviewer for suggesting this expansion of our text. Indeed, inflammaging might be associated with certain mechanisms of DNA repair. We have added the relevant information and reference (lines 108-120).

Chapter 3.1: I will add a caveat on the tissue type where the DNA methylation age has been measured (usually blood cells), which does not necessarily reflect the trend of other cell types that may be more relevant for longevity (i.e. brain, liver, stem cells).

        Indeed one has to be careful when generalizing results of tests (like DNA methylation) which were  measured in certain tissue(s) to other tissues. We thank the reviewer for pointing this out. We have added the requested caveat (lines 160-165).

Chapter 3.2: Ref. n. 51 is not correctly reported, please rephrase.

          We thank the reviewer for pointing out our mistake. We have rephrased the relevant text (lines 178-183); the reference changed its number to [52] due to addition of another one before.

Chapter 4: the part on mTOR is oversimplified, since mTOR is not merely a protein whose function can be decreased or increased, but it is rather a hub where an integration of signals occurs leading to a decision on cell fate. Moreover, from an evolutive point of view, if the authors were right, it is more than likely that hypofunctional variants of mTOR would have been eliminated by natural selection. Please reconsider this section.

         We agree with the reviewer that part of our text concerning the mTOR was oversimplified. We have now expanded it, added 2 new references and hope it is now adequate (lines 277-287).

Chapter 5: please explain more in detail the hypothesis of mitohormesis. A brief mention should also be done to the fact that a little mitochondrial dysfunction can be good for longevity as it can elicit a beneficial stress response.

          Indeed, mitohormesis may play role in longevity. We have  expanded on this subject in lines 388-399 of the revised text, adding 3 new relevant references.

Chapter 6: a more detailed discussion can be provided for N-glycosylation (e.g. PMID: 22353383; PMID: 18047421).

     We thank the reviewer for this remark We have expanded information on N-glycosylation and its role in aging and aging-related diseases (lines 410-424). We have added  suggested references and onother 2 that we found while working on revision of our text.

Chapter 8: an authors’ expert point of view would be desirable. Would the authors mind to speculate on the geroproteome? What are the causes for its onset? Have centenarians always been exceptionally gifted in terms of proteome maintenance, or they adapted better than others to environmental situations? At least in hypothetical form, a guess on this would be welcome.

         We thank the reviewer for  these questions. Indeed answering them is so far a speculation. Still, based on available information we believe that the causes of proteome NOT becoming a geroproteome are both genetic (being an "exceptional gift") and   related to better overall adaptation of centenarians to environmental and other stresses and insults. To this end we have added lines 672-679.

Minor points: line 46: a series of 3 full stops is present; Chapter 8: the word “conclusions” contains a typo.  

    We have corrected the indicated typografical errors and a few more that we have found while preparing our revised text.

   We hope our explanations given here and first of all in the revised text will be sufficient.

Reviewer 2 Report

It is a very interesting manuscript that resumes the recent findings about the impact of cellular proteodynamics on longevity. The authors describe the mechanisms controlling the cellular proteodynamics at molecular and cellular levels including DNA, mRNA, miRNA, lncRNA, tRNA, ribosome. They focused also on the role of proteostatic systems and systems responsible for protein degradation including autophagy, metalloproteinases (ADAM) and ADAM with thrombospondin domain (ADAMTS), calpain-calpastatin system. Because the authors described many types of RNA and their roles in proteodynamics, I recommend to provide an additional section about posttranscriptional modifications of RNA (m6A, m5C). The latest publications bring a new knowledge about significance of above processes on aging and proetodynamics (see. DOI: 10.1111/acel.13657, https://doi.org/10.3389/fgene.2022.869950, or doi: 10.1002/wrna.1547). What is known about the role of m6A, m5C during regulation of the activity of protein synthesis/degradation systems? The authors should provide information whether the lack of enzymes responsible for RNA methylation disturbs the process of protein synthesis and degradation. I suggest adding this information e.g., as a table.

Author Response

Because the authors described many types of RNA and their roles in proteodynamics, I recommend to provide an additional section about posttranscriptional modifications of RNA (m6A, m5C). The latest publications bring a new knowledge about significance of above processes on aging and proetodynamics (see. DOI: 10.1111/acel.13657, https://doi.org/10.3389/fgene.2022.869950, or doi: 10.1002/wrna.1547). What is known about the role of m6A, m5C during regulation of the activity of protein synthesis/degradation systems? The authors should provide information whether the lack of enzymes responsible for RNA methylation disturbs the process of protein synthesis and degradation.

    We thank the reviewer very much  for pointing to us this important ommission in our ext. Indeed, numerous types of modification of different species of RNA exist and may affect the proteodynamics both in beneficial and detrimental way, participating in longevity or in aging-related diseases' pathomechanisms. To this end we have added, as suggested, a sectiion 3.3 (lines 185-243 of revised manuscript) and added 11 relevant references. We believe this expansion of our text would make it more comprensive and hope it will be accepted by the reviewer.

Reviewer 3 Report

This is an interesting and updated review on a relevant biological matter. The authors have a good background of previous publications on this field. Although it can be accepted as is, I find that this review is merely descriptive as the changes of biological markers and mechanisms of proteodynamics are commented only in qualitative terms. This is somewhat misleading, because it gives the same relevance to minor (secondary) and to major (main) changes of biological markers. Due to this, the conclusions drawn are only generic statements. Therefore, I suggest to improve this review highlighting the proteodynamics markers for which have been reported the largest alterations, including available quantitative data, for example, n-fold increase or percentage of decrease.

The English needs minor corrections. For example, the heading “CONLUSIONS” should read “CONCLUSIONS”.

Author Response

e thank the reviewer for their comment on our work. We agree, that  our "review is merely descriptive as the changes of biological markers and mechanisms of proteodynamics are commented only in qualitative terms" and " gives the same relevance to minor (secondary) and to major (main) changes of biological markers". The reviewer suggests "to improve this review highlighting the proteodynamics markers for which have been reported the largest alterations, including available quantitative data, for example, n-fold increase or percentage of decrease. " However, such task is in our opinion practically impossible to be fulfilled, as data on different proteodynamics markers in centenarians (which is the topic of our text) are usually given in just a single original paper. Thus, it is hard to define which changes are major and which are minor; even if some numerical values could be given for these changes, they cannot be compared among many reports on the same proteodynamic biomarker, as such multiple reports simply do not exist. On some occasions (brought up in our text) certain aspects of centenarian proteodynamics  were studied by more than one group in subjects of different ethnicities (e.g., Italian and Japanese or Chinese), but this is exceptional. We know that this response may not be satisfactory for the reviewer, but  we would need many more published data in order to be able to choose the most significant, major proteodynamic biomarkers characterizing the centenarians and being most important for maintenance of healthy proteomes even at that advanced age.

Round 2

Reviewer 1 Report

The review has been extensively revised and all the issues and suggestions raised by this Reviewer have been satisfactorily addressed. 

No further comments.